# Digital Technologies in the Architecture, Engineering and Construction (AEC) Industry—A Bibliometric—Qualitative Literature Review of Research Activities

**DOI:** 10.3390/ijerph18116135

**Published:** 2021-06-06

**Authors:** Bilal Manzoor, Idris Othman, Juan Carlos Pomares

**Affiliations:** 1Department of Civil & Environmental Engineering, University Technology PETRONAS, Seri Iskandar 32610, Perak, Malaysia; bilal_18003504@utp.edu.my (B.M.); Idris_othman@utp.edu.my (I.O.); 2Civil Engineering Department, University of Alicante, P.O. Box 99, E-03690 Alicante, Spain

**Keywords:** digital technologies, bibliometric analysis, systematic review, building information modeling (BIM), safety, accidents

## Abstract

Digital technologies (DTs) are proven helpful in the Architecture, Engineering and Construction (AEC) industry due to their varied benefits to project stakeholders, such as enhanced visualization, better data sharing, reduction in building waste, increased productivity, sustainable performance and safety improvement. Therefore, researchers have conducted various studies on DTs in the AEC industry over the year; however, this study explores the state-of-the-art research on DTs in the AEC industry by means of a bibliometric-qualitative review method. This research would uncover new knowledge gaps and practical needs in the domain of DTs in the AEC industry. In addition, bibliometric analysis was carried out by utilizing academic publications from Scopus (i.e., 11,047 publications for the AEC industry, 1956 for DTs and 1778 for DTs in the AEC industry). Furthermore, a qualitative review was further conducted on 200 screened selected research publications in the domain of DTs. This study brings attention to the body of knowledge by envisioning trends and patterns by defining key research interests, journals, countries, new advancements, challenges, negative attitudes and future directions towards DTs in the AEC industry. However, this study is the first in its vital importance and uniqueness by providing a broad updated review of DTs in the AEC literature. Furthermore, this research laid a foundation for future researchers, policy makers and practitioners to explore the limitations in future research.

## 1. Introduction

The Architecture, Engineering and Construction (AEC) industry is undergoing a significant shift from conventional labor-intensive methods to automation through the use of digital technologies (DTs) and has played a significant role in this revolution [1]. In general, DTs generally refer to information and communication technologies (ICTs) that facilitate the development, storage and handling of information and promote the various forms of communication between human beings and electronic systems and between electronic systems in digital binary computing systems [2]. With the great emphasis on these technologies in the AEC industry, the assessment of their impact on user behavior is becoming increasingly important as DTs transform the relationship between the construction process and human behavior [3]. As a result of growth in the concepts of digitalization and automation in Industry 4.0, DTs have gained rising attention [4,5]. They could be used for logistics operations, near-real-time knowledge flows, end-to-end supply chain consistency and increased human interaction through the application of digital technology, in particular labor-intensive activities [6].

Building information modeling (BIM) [7], augmented reality (AR) [8], virtual reality (VR) [9], photogrammetry [10], radio frequency identification devices (RFID) [11], geographic information systems (GIS) [12], global positioning systems (GPS) [13], wearable safety devices [14], quick response code (QR) [15], artificial intelligence (AI) [16], robotics [17], block chain [18], onsite mobile devices [19] and laser scanning devices [20] are all seen as promising DTs in the AEC industry. In general, the advantages of using DTs in the AEC industry are known to be immense. For instance, AR-VR technologies have shown promising results in a number of fields related to the AEC industry offering solutions to collaborative and communication issues [21]. Methodologies such as BIM for project documentation and architecture management enable teams to work on a cooperative network, collaborate and share project data [21]. Various examples of VR integrations assembled for applications associated with the AEC show promising results, e.g., urban planning [22], acoustic analysis [23], design analysis and decision-making support [24], construction safety [25], enhancement in spatial perception and learning results in engineering related areas [26]. Furthermore, RFID can help boost the transparency and productivity of the supply chain [27]. Likewise, for data collection, tracking, visual monitoring and assessment purposes, GPS–GIS technologies can be used [28]. While robotics use has the potential to increase efficiency and safety, it should not inherently in the longer term reduce total work opportunities in the construction sector [29]. Using the camera of a smartphone, an application reads the QR code and shows the clash-free model on its display. Users can incorporate, interpret or use section planes to easily visualize the design of the drawings [30]. In addition, AI offers huge potential for substantial efficiency improvements through rapid and accurate analysis of large data volumes [31]. In addition, AI systems and technologies can resolve complex, non-linear functional issues and once equipped, predictions and generalizations can be carried out at high speed [32].

With the increasing demand for DTs in the AEC industry, more studies are needed in this field. However, previous review studies in this field [16,33,34,35] have made significant contributions but have various limitations. Firstly, they have been used manual methods [36]. Secondly, there is a lack of comprehensive reviews of state-of-the-art research covering all the aspects of DTs under one umbrella. For instance, Ibem and Laryea [35] conducted a literature review that only focus on the procurement of construction projects. Guo et al. [34] conducted a literature review but only limited to the boundary of construction safety. Pauwels et al. [27] conducted a literature review but limited to the benefits of semantic web technologies. Darko et al. [16] conducted the first comprehensive scientometric study assessing the state-of-the-art of research on AI in the AEC industry, but limited within the domain of Al. In the context of this evidence, these analysis studies do not provide a complete picture of state-of-the-art research on DTs in the AEC industry.

In fact, there is still a lack of a study that provides a detailed image and understanding of the literature of DTs in the specific domain of AEC. Therefore, the objectives of this study are: RQ1. What is the annual publications trend on DTs in the AEC industry? RQ2. Which journals publish studies on the contribution of DTs in the AEC industry? RQ3. In which context (country) were the studies conducted? RQ4. Which digital technology emerges the most among all other DTs? RQ5. What are the new advancements, challenges, negative attitudes and future directions towards DTs in the AEC industry?

In order to overcome this research gap, a quantitative (i.e., bibliometric analysis) and qualitative approach (i.e., systematic review) were used to seek responses to these questions for the benefits of academia and industry. It is anticipated that the answers to these questions would give rise to contributing knowledge in literature and DTs. As a consequence of DTs are gaining broader market development and the interest of practitioners, construction stakeholders and researchers in the global environment to benefit from new advancements and future directions.

## 2. Background of Digital Technologies

Four phases in the evolution of DTs have been addressed in literature studies. The first step was pre-1838 when Samuel Morse invented the first telegraphic transmission and manually transmitted information. Following this, electricity and electro-mechanical control were invented, and gadgets such as the telegraph, telephone, radio, television were developed. The third step was the convergence, separately used, of broadcast and electronic technology in the 1950s and data were transmitted in analogue mode before the 1950s. The final step was to replace the analogue mode of data transfer with the digital system [37]. In the 1960s, electronic devices and signal processing were introduced and gradually accelerated as digital goods such as CDs were introduced into the market in the 1980s. It is evident from this that ICTs were incubated for more than 100 years, while their digitalization, i.e., the process of translation into electronic and binary machine language by sound, text, voice or image, was taken 20 years and so on to execute [37].

In particular, the digitalization of ICTs increases the ability of network technologies, the performance of data and information exchange and cost prosperity in the storage, transmission and processing of information [38]. The ability of DTs to integrate electronic networking and computing in data processing and manage various data sorts simultaneously (e.g., text, sound or image) has thus led to the provision of computer equipment, telecommunications systems and networks that have problems [39]. Thus, the three basic characteristics of DTs: integration and multi-functionality; intelligence and prevalence are undoubtedly very important features; where good data processing, powerful connectivity of tasks and procedures, clear communication, coordination and collaboration of the workflow are needed [40].

In addition, DTs have various applications such as communication, computing accessories, networking gadgets, intangible items, software applications, communication networks and the Internet. DTs can also promote the execution of a wide range of enterprise and natural processes, including manufacturing. [41]. For their applications, therefore, DTs can be divided into six key classes. These include data and information capture, storage, processing, collaboration and display, as well as for integration, collaboration and synchronization of work processes with storage, collaboration and interoperability service abilities; for example, keyboards, floppy discs, hard discs, mobile devices, printers, fax machines, readers bar code and scanning images [42].

Since its inception, the AEC industry has pursued ways to reduce costs, improve efficiency, enhance visualization, improve data sharing, reduce building waste, increase productivity, sustainable performance, improve safety and boost quality, all while attempting to reduce delivery time. The AEC industry is still heavily reliant on conventional drawings and procedures in order to conduct business [43]. Simultaneously, the AEC professionals recognize the value of DTs in achieving more precise and intelligent modeling. Since the AEC industry is highly competitive, those AEC firms that effectively implement the latest technologies will be able to outperform the competition due to their ability to adopt them. Despite enormous automation opportunities, the AEC sector has only recently begun the transition from tradition to automation [15].

## 3. Research Methodology

To fulfill the research objectives, a “mixed-systematic review method” was used in this study and comprises of a quantitative approach (i.e., bibliometric analysis) and a qualitative approach (i.e., systematic review). The flowchart of research methodology is shown in Figure 1.

### 3.1. Mixed-Systematic Review Method

Quantitative and qualitative approaches are collectively referred to as a mixed-systematic review method. The purpose of the mixed approach seeks to use the power of qualitative and quantitative approaches and to reduce the inconvenience [44]. To imagine the systemic and complex dimensions of scientific research, the bibliometric approach is used [45]. Bibliometric visualization is an important tool that visualizes the domain of information and the relationship among papers, journals, and so on [46]. In this study, bibliometric mapping is used to classify knowledge domains and research patterns for DTs in the AEC industry based on established literature. Meanwhile, a systematic analysis is underway to include a holistic view of current research to assess lacunae in the knowledge pool and predict future research directions [47].

### 3.2. Data Collection

The data acquisition has been conducted with the aid of Scopus rather than Google scholar or web of science due to the following reasons (a) Scopus has a broad range of scientific journal publications relative to other databases [48], (b) Scopus has a much greater indexing mechanism that improves the opportunity to access more recent publications [49], (c) previous studies [16,50] used Scopus instead of other databases due to avoid from the difficulty of checking and replication of publications from various databases. The data collection in this study was achieved by two approaches (a) bibliometric analysis and (b) systematic analysis. The bibliometric analysis provides a platform for deeper insight selection of data regarding DTs in the AEC industry. Table 1 elaborates the searching strategies of literature. The first and second queries in Table 1 were merged in order to retrieve existing literature related to DTs in the domain of the AEC industry. “Digital” OR “technology” OR “technologies” AND “Architectural engineering” OR “Construction engineering” OR “Civil engineering” AND “BIM” OR “Building information modeling” OR “Building information modelling” OR “AR” OR “VR” OR “RFID” OR “Photogrammetry” OR “Images” OR “Videos” OR “Cameras” OR “laser scanning” OR “3D scanning” OR “Robotics” OR “Block chain”.

The Scopus search keyword was set as title/abstract/keywords to extract all publications in their title, abstract or keyword section. In this search, 1778 Scopus records were retrieved. A screening technique was successively conducted with the help of a systematic review in order to boost the findings to the applicable engineering scope. Study papers other than engineering, including medicine and agriculture, were additionally omitted. The source title was analyzed to extract irrelevant papers and conference proceedings by performing another refining. The reason to exclude conference papers is that conference papers did not go through the peer-reviewed process before publication. Now, the number reduced to 445 Scopus records. Further, refining was conducted to read the title and abstract of each individual paper and exclude the irrelevant papers. For example, the papers that were outside the domain of DTs and AEC boundaries and non-English publications. The final outcome was 200 articles from 1975 to December 2020.

### 3.3. Bibliometric Analysis Tools

There are various bibliometric analysis tools such as Gephi, CiteSpace, Sci2, HistCite and VOSviewer available, each of which has its own potential and abilities. In this study, VOSviewer was used that provides the basic functionality needed to produce, visualize and explore bibliometric networks [51]. For instance, Vilutiene et al. [52] used the VOSviewer tool for bibliometric analysis of BIM for structural engineering. In addition, Santos et al. [53] used the VOSviewer tool for bibliometric analysis and review of BIM literature. The most recent study conducted by Babalola et al. [54] also emphasized the usage of VOSviewer in bibliometric analysis of advances in BIM.

## 4. Results and Discussion

### 4.1. Annual Publications Trend on DTs in the AEC Industry (20th and 21st Century)

As compared between the 20th century (1975 to 2000) and the 21st century (2001–2020), the trend of publications on DTs in the AEC industry is high in the 21st century (2001–2020). This is a fact by Robin [55] that DTs have become the most powerful technology tool in the 21st century. Bilal et al. [56] also provide evidence of an increasing trend of DTs in the AEC industry in the 21st century. In the 20th century, during the late 1990s, research in DTs had gained momentum when computational capacity and researchers’ dedication to using DTs in resolving more complex issues in a wide range of fields began to increase [57]. Due to this, the number of publications might had caused a little peak in 1996. However, the trend of publications on DTs in the AEC industry was boosted up in 2014–2020. This increase in the trend of publication may be due to the need and level of adoption in the AEC industry [58]. It shows that the adoption of DTs in the 21st century is increasing and raising the benefits. For example, compared to the 20th century, it is difficult to visualize the construction site. However, photogrammetry has made visualization simpler and quicker in the 21st century. Similarly, the monitoring process of the construction area is made reliable with the help of DTs. Figure 2 depicts the comprehensive annual pattern of publication in the domain of DTs in the AEC industry.

### 4.2. Journal Contributions on DTs in the AEC Industry

Numerous studies have shown and made clear how important academic journals are to be examined in any field of science [59]. The results suggested that the most important contribution of publications on DTs in the AEC industry by the *Automation in Construction* followed by *Journal of Construction Engineering and Management* and *Journal of Computing in Civil Engineering*. This indicates that practitioners, researchers and students who wish to undertake more research in the domain of DTs in the AEC industry should place a greater priority on *Automation in Construction* due to its significant contribution in the field of DTs in the AEC industry. The mission of this publishing initiative is to advance the subject through publications that include a wide range of information on all aspects of information technology application in the areas of design, engineering, construction and maintenance and administration of developed facilities. To put it another way, it deals with issues such as automated monitoring, smart control systems, computer-aided design and engineering, etc. The contribution of *Automation in Construction* indicates the most advanced research in the domain of digital construction. Keeping this in mind, this will help future researchers to explore more in the field of digitalization construction and to improve safety performance. However, it is worth mentioning that aside from *Automation in Construction*, which has the highest contributions, all the others are the AEC industry journals, though the *Journal of Cleaner Production* and some of the other Engineering journals (*Composite Structures, Buildings, Journal of infrastructure systems, Visualization in Engineering, Construction Innovation, Accident Analysis and Prevention*) may include vast area of engineering. The detailed theme of journal contributions on DTs in the AEC industry is shown in Figure 3.

### 4.3. Bibliometric Analysis

#### 4.3.1. Geospatial Distribution of Research Articles on DTs in the AEC Industry

The network of countries in the field of scientific collaboration helps to identify countries that are particularly interested in the field of related research. A network was developed with the VOSviewer for geospatial distribution and collaboration. This type of analysis is known as “co-authorship”, the unit of analysis is known as “countries”. Figure 4 shows a detailed picture of the countries’ collaboration network.

To achieve the optimum network, the “minimum number of country documents” and the “minimum number of country citations” were both set to 15. It can be seen that the U.S. and China have a strong bond with each other and stand at the top-ranked countries in research on DTs in the AEC industry, followed by Spain and the United Kingdom. Furthermore, the U.S. has also strong collaboration and the biggest contributor in research related to DTs with Germany, Greece, Hong Kong, Canada, South Korea and Israel. The Table 2. illustrates the detailed picture of geospatial distribution of research articles on DTs in the AEC industry.

#### 4.3.2. Author Keywords Co-Occurrence Analysis

With the aid of VOSviewer, it is possible to build up the knowledge domain of DTs in the AEC industry by using author keywords co-occurrence analysis by using 200 selected papers. The performance of the VOSviewer (i.e., Network Visualization) is a distance-based map in which the spacing objects represents the strength of the relationship between the items [46]. The representation of various groups of items is shown by different colors that are clustered by the clustering technique of VOSviewer [60]. The node sizes demonstrate the occurrence frequency of the relevant keywords, the arcs simply show the co-occurrence relationship between the keywords, and the line thickness indicates the strength of each relationship [61,62,63].

Various studies also support the use of author keywords for bibliometric analysis [53,64,65,66,67]. Most recent studies such as BIM for structural engineering [52], construction education [68], BIM-based research in construction engineering [69], mapping the knowledge domain of BIM [70] and integrating BIM with building performance [71] also had the recommendation to use this approach. Furthermore, identical terms such as BIM, Building information modeling, Building information modelling and information technologies, information technology, technology were merged as BIM and information technology, respectively. The detailed analysis of keywords with occurrences, average publication years, links and total links is elaborated in Table 3.

Average publication year means average time period in which specific research areas have received attention from the researchers. For instance, photogrammetry, computer vision, terrestrial laser scanning, buildings, displacement measurement and education gained attention in 2016, while studies focused on BIM, civil engineering, unmanned aerial vehicle, case study, LiDAR and structural health monitoring gained attention in 2017. The “links” are the number of connections between a particular items and another, while the “total links strength” represents the total strength associated with particular item [46,72].

There are six clusters omitted on the basis of Figure 5 which are summarized as follow:(i)BIM Integrating Cluster (Green Color)

In this cluster, BIM is integrated with architectural engineering, text mining, case study, education, and curriculum. BIM has the potential to reduce rework for various engineering goals and to enhance decision-making in different aspects in the AEC industry [73]. Various studies provide the recommendation of BIM research using text mining techniques [74,75,76]. A recent study conducted by Pan and Zhang [77] for mining BIM log data with the approach of Long Short-Term Memory Neural Network (LSTM NN). Although this research had major contributions but had a limitation to implement it as an Autodesk Revit plugin for a better user experience. Furthermore, to achieve tremendous achievements and new mindsets, the emerging technology BIM needs more education practices [78]. Universities must concentrate on the strategy of using BIM as a creative technology to encourage students to learn new skills and prepare them for their future activities in a more competitive environment [79]. However, very limited studies had been conducted on strategies implemented by educators to promote BIM education, which is not only beneficial for the AEC industry but also for BIM experts, developers and researchers [80,81]. Therefore, more proactive collaborations and BIM courses in civil engineering curriculum design are required to advance BIM educators and BIM talent development in the AEC industry [82,83].

(ii)Automation Integrating Cluster (Yellow Color)

The automation cluster is integrated with construction, information technology, visualization and artificial intelligence. The concept of automation began in the last decade with the aim of reducing manpower and time [84]. The development of AI is speeding up rapidly, and the combination of AI with automation has started to change the AEC industry [85]. To achieve the new norm of quality and advancement in construction tasks, the AEC industry is adopting AI with an automation process [86]. Automation has implemented a framework for computers and devices and replaced the mechanism that was designed by combining the man with the computer [87]. In addition, ‘construction automation’ is a set of a new generation of technological innovations that will radically change the entire process and philosophy of construction [87]. Basically, executing the construction activities with the aid of robots is termed ‘construction automation’ [88]. Some researchers claim that ‘construction automation’ is the combination of computer-aided design and robot-based on-site techniques to accelerate overall activities [89]. Some of them, such as Willmann et al. [90], now use the term ‘digital manufacturing’ as a synonym for ‘construction automation’, especially when it comes to designing building projects. It is recommended that construction automation is actually occurring broadly nowadays; it is vitally important from the outset to build a standard practice to make clear the significant benefits and potential outcomes through data-based approaches and to establish a thorough understanding of construction automation [91].

(iii)Photogrammetry Integrating Cluster (Pink Color)

The cluster of photogrammetry comprises (light detection and ranging) LiDAR, terrestrial laser scanning, buildings, damage, mapping, point clouds and inspection. In the last two decades, 3D building modeling became one of the most popular and hottest topics in photogrammetry, and it seems that photogrammetry is the only commercial way to acquire genuine 3D city data [92,93]. The majority of methods for 3D building modeling and LIDAR for building structures can be improved by oblique models so that the simulation of the facade or other manual reconstruction processes is only required for the user so that the operator can decide the boundaries to reflect the building model, and the process is not too time-consuming [94,95]. Due to terrestrial laser scanning results, the reconstruction of the facade has proven to be a valuable source in recent years. There can be up to 100 or 1000 points per square meter of steady laser scanning density in urban areas, which is high enough to record a lot of data on building facades. Various measures, such as image matching, intersection and resection, can be skipped compared to image models, whereas image interpretations are not needed for laser-based reconstruction approaches that are faced with major challenges such as extracting significant structures from large quantities of data [96]. Furthermore, a recent study conducted by L. Yang et al. [97] explored the semi-automated generation for steel structures based on terrestrial laser scanning data. It is recommended to expand the established technique to other popular types of project structures, such as L-shaped structures, T-shaped structures and other structures generated by the combination of simple primitives.

(iv)Civil Engineering Integrating Cluster (Blue Color)

The civil engineering cluster is the combination of digital image correlation, displacement measurement, structural health monitoring, computer vision and unmanned aerial vehicle. Civil engineering infrastructure such as bridges, houses and tunnels continue to be used amid the ageing and corrosion of their construction life [98]. Conventional inspection and monitoring techniques can yield contradictory results, are labor-intensive and too time-consuming to be considered successful for large-scale monitoring [99]. New structural health monitoring systems must therefore be developed that are automated, highly accurate, minimally invasive and cost-effective [100,101]. Three-dimensional (3D) digital image correlation (DIC) systems are capable of extracting full-field strain, displacement, and geometry profiles [102]. Furthermore, when this measurement technique is introduced within the Unmanned Aerial Vehicle (UAV), the ability to accelerate the optical-based measurement process is improved as well as the downtime of the infrastructure is minimized [103]. These resulting credibility maps of the interest framework can be easily interpreted by qualified staff [104]. Moreover, a recent study conducted by Khuc et al. [105] attempted to enhance the displacement measurement methods by incorporating UAV and computer vision algorithms. Although this research has many advantages but also have some limitation such as UAV cameras, weather conditions, UAV crashes and turbulence. These limitations should be feasible for future works [105].

(v)Augmented Reality (AR) and Computer Applications Integrating Cluster (Light Blue Color–Orange Color)

AR integrated cluster consists of the construction site and mobile computing and represented by light blue in color, while computer applications cluster consists of tracking and represented by orange color. The concept of using AR technology for building planning has gained popularity as desktop computers have supported more sophisticated graphics capabilities [106]. It has been found that AR is supposed to be as large a step as the transformation from 2D line drawings to photorealistic 3D projections [107]. In addition, the use of mobile computing in the building is becoming an important research subject in the field of construction information technology [108]. Although AR and computer applications play a tremendous role and boost up the DTs, they have some limitations [109,110]. It is suggested to integrate AR with other emerging digital technologies such as block chain, IoT (internet of things) and MR (mixed reality) to create a near-real-time virtual environment [111].

(vi)Construction Management Integrating Cluster (Purple Color)

Construction management integrating cluster consists of construction education, construction technology and RFID. Various studies have been conducted on the influence of RFID on construction automated management [112,113,114]. Furthermore, the incorporation of RFID and BIM significantly enhances quality control, work logistics, construction safety and has become a hot topic in related studies [115,116,117]. However, there is a need to further integrate RFID with other emerging DTs in the AEC industry for better visualization in construction management activities. The key challenge of raising the productivity of RFID and its incorporation with other digital technologies (e.g., GIS and AR/VR) is the lack of interoperability [118,119,120] concerning data storage, storage, semantics and ontology [121,122]. For instance, Xie et al. [123] defined one of the key semantic interoperability gaps between RFID and VR, needing further future work to improve the interoperable framework.

### 4.4. Digital Technologies in the AEC Industry

Based on 200 papers, it was discovered that BIM is the most important digital technology (DT) in the AEC industry, followed by robotics, photogrammetry, laser scanning, images, AR, image processing techniques, RFID, VR and mobiles. As shown in Figure 6, BIM has a greater presence in the AEC industry than other DTs. BIM provides the possibility of simulating a construction project in a simulated environment. A building information model, or an accurate virtual model of a building, is digitally built using BIM technology. When finished, the building information model will contain detailed geometry and relevant data to support the design, procurement, manufacturing and construction activities necessary to realize the building. Although every DTs have its own credibility however various researchers used BIM integration with other DTs due to the more potential in the AEC industry. For instance, BIM integration with RFID [114,124,125,126], BIM integration with AR/VR technologies [127,128], BIM integration with IoT [129,130,131,132], BIM integration with GIS [65,133], BIM integration with laser scanning [134,135,136] and BIM integration with UAV technology [10,137]. Robotic technology has also been found to be the second most emerging technology after BIM, as shown in Figure 6. Robotic technology has the potential to reduce labor costs and improve quality and productivity [138]. Furthermore, robotic technology can reduce damage and free employees from hazardous tasks [138]. Bock [87] argues that traditional construction approaches have reached their limits and that robotics and automation technologies are capable of addressing the challenges of efficiency in the AEC industry. The history of robotic technology was developed in the 1960s, but the level of adoption of robotic technology in the AEC industry is very slow [139]. Several studies have highlighted the factors leading to the lower adoption of robotic technology. For example, a detailed study was presented on the challenges of robotics development in Japan, Malaysia and Australia [140]. Although there is considerable potential for robotic technology in the AEC industry, the challenges in the adoption of robotic technology have not been fully documented. It is, therefore, recommended that there be a clear, up-to-date study and further discussion of limiting adaptation factors [141].

### 4.5. Systematic Analysis

The systematic analysis of selected papers was provided in this section in order to provide an in-depth insight into research on DTs in the AEC industry. Among the 1778 publications resulting from bibliometric analysis, a screening procedure was performed to remove duplicate articles, irrelevant papers, and non-peer-reviewed papers. In the end, 200 articles were chosen following the screening process. Based on the subsequent papers, new advancements, challenges, negative attitudes and future directions have been explored.

#### 4.5.1. New Advancements towards DTs in the AEC Industry

The phenomenon of DTs has been emerging since the 1930s, and with the passing of time, new technologies, possibilities and growth have been seen in the AEC industry [142]. Compared to the 20th century, the 21st century is well ahead in terms of technology, innovation and digitalization, and is also on the rise [55,143]. For example, BIM is the latest and emerging DT of the 21st century; no doubt the BIM idea was developed in the 20th century [144], but BIM reached its height in the 21st century [145,146]. Thus, BIM is a growing area of study and practice that integrates the diverse information domains of the AEC industry [147]. That is why other DTs have been integrated with BIM, such as BIM–RFID [124,125], BIM–GIS [148], BIM–AR/VR [149,150] and BIM–UAV [137,151] and so on. In addition, BIM was presented as a major technological improvement on conventional CAD, providing more intelligence and interoperability capabilities [152]. It was found that there several studies related to BIM integrates with sustainability and green house buildings [153,154,155,156,157,158] in developed countries. A recent study conducted by Naji et al. [159] explored the BIM in lower consumption electric power but was limited to questionnaire data and simulation work and provide a platform for future researchers to explore this simulation work in real-time case studies for validation purposes. Now, it is time to explore more energy consumption techniques, new carbon emission techniques integrating with BIM and other emerging technologies in developing countries [160].

Furthermore, it was found that very few studies have been conducted on BIM integration with block chains [161]. Hence, there is a need for time to explore more studies on block chain. Block chain is basically a growing emerging technology that has received significant momentum in various industries in both the public and private sectors in recent years [162]. With the advancement of block chain technology, the direction has gained new momentum in the smart contract technologies such as chain codes, automated code-checking compliances and BIM–cyber security to provide a platform for transparency and security [163]. Therefore, it is recommended to explore more research in the field of block chain and cyber security incorporated with BIM will gain a new boost in the AEC industry. Table 4 elaborates the new advancements towards DTs in the AEC industry.

In addition, the integration of DTs with safety management was explored in previous studies [184,185]. In developed countries such as Spain, Germany and the United Kingdom, BIM has already been introduced in the area of construction safety management [186]. Future research in developing countries with the incorporation of BIM with other DTs in the field of safety management needs to be discussed [117].

Last but not least, mobile technology leads towards DTs in the AEC industry in a bright way. With the development of 5G technology, there is a boost in the field of DTs [187]. A recent study conducted by Chew et al. [188] explored the roadmap of 5G technology for smart buildings. However, this work has various contributions in the AEC industry but is limited to building structures. This work can be extended to other infrastructures as well. In addition, a recent survey of 6G exploration was conducted in China [189]. With the advancement of 6G technology, the AEC industry will be improved, and thus, the DTs will have a significant effect on the growth and prosperity prospects of the AEC industry.

#### 4.5.2. Challenges towards DTs in the AEC Industry

In this section, several challenges have been explored for the adoption of DTs in the AEC industry. However, it was concluded that more than 80% of the cost of using technology is estimated to occur after the initial purchase of the technology [190]. Purchasing a VR and AR headset with complete features for construction use could cost as little as $500 [191]. However, support systems cost, such as game engine software, and hardware components cost, such as laptop, phone and motion tracker, could easily surpass $5000. In addition, to build an interactive testing environment for the significant use of on-site DTs, the developer should be employed, and this requires huge costs [191]. The same applies to the use of WSDs, and these devices are cheap to acquire but very costly to use and maintain for a specific period of time. It was found that IoT-supported wearable devices could cost $100 per clip-on unit, with an additional networking cost of $12,000–24,000 per year [192]. Besides costs, there are various challenges that are well documented in Table 5.

It has been found that numerous challenges are on the way to reducing the adoption of DTs in the AEC industry. However, several researchers have conducted a study to find a solution and alleviate the challenges of implementing DTs in the AEC industry, but more work still needs to be conducted. Future work needs to be conducted on strategies to reduce challenges in order to smoothly implement DTs in the AEC industry.

#### 4.5.3. Negative Attitudes towards DTs in the AEC Industry

In this section, negative attitude towards DTs in the AEC industry has been explored. Corruption is the most dominant factor leading the AEC industry to decline [214]. The negativity of corruption exists at every point of construction projects, including numerous project stakeholders, government officials, engineers, suppliers and so on [215]. For a significant time, the AEC industry had no concerted strategy to tackle the issue, but a series of events occurred in the late 1990s that contributed to the industry’s most far-reaching attempts to counter corruption. Global governments, engineering/construction organizations and individuals are making efforts to eradicate corruption and conduct business in an honest, transparent and equitable manner [215]. In addition, a range of other negative attitudes towards DTs in the AEC industry include kickbacks, bribery, tender rigging, fraud, unethical practices and conflicts of interest [216]. Furthermore, kickbacks and bribery are considered as the two faces of the same illegal coin. However, tender rigging is another crucial problem that can take a number of forms. Owners’ employees can participate by setting a very short period of bidding so that only companies they have unlawfully informed of the forthcoming offer have adequate time to prepare a sound bid. Owners’ workers can also exempt interested firms from the bid list and only allow “favored” firms to participate [217]. Likewise, fraud, unethical practices and conflict of interest also produce negativity on the way of development and productivity. Fraud is an economic crime involving activities such as swindling, trickery, misinformation or deceit [218]. Hence, it can be said that these negative factors not only harm the image of the AEC industry but also pose serious questions about DTs. Table 6 illustrates a better picture of negative attitudes towards DTs in the AEC industry.

#### 4.5.4. Future Directions towards DTs in the AEC Industry

Robotics is considered to be one of the primary fields for DTs in the AEC industry, but little attention has been paid to robotic systems. Performing various construction tasks is dangerous and can lead to serious injuries and deaths, such as construction works above the sea, at heights and inside deep trenches; thus, there is a need to develop user-friendly, advanced and smart robots in future. The majority of the recent studies focused on the use of AI methods to define, assess, monitor and handle security threats instead of replacing people with robots in risky conditions. It is really important to concentrate future research on inventing robots that function without human interference to solve that problem [228]. In addition, it is suggested that collaborative robots be created and used, i.e., robots designed to work with humans as employees add value to projects instead of replacing people fully with robots. Robotics would add immense value to the AEC industry, such as improved efficiency, performance, safety and quality. Likewise, in the AEC industry, modularization and pre-fabrication, along with 3D printing, these advantages are now becoming a prevalent approach. Another interesting area of science, such as the development and implementation of robots for building construction in factories, is the integration of robotics with modular construction technology and 3D printing technology. The key question here is how robots can affect employee emotions and performance. This should be addressed in future studies. It is also recommended to integrate BIM with cyber security to ensure the potential of security of databases.

## 5. Summary of Findings

Figure 7 provides a good picture of the overview of results in DTs in the AEC industry under one umbrella. It is clear that the United States has the leading research in the field of DTs led by China, Spain, the United Kingdom and Australia. The main topics of the author’s keyword co-occurrence are also related to BIM, construction management, civil engineering, photogrammetry, construction, AR, information technology and automation. In addition, BIM is considered to be the most emerging DTs among all other DTs in the AEC industry. BIM–block chain integration is seen as the most exciting new technology in the AEC industry in terms of DTs. In addition, transparency and security issues need to be further discussed in the future. In addition, inadequate skills, government policies, barriers to culture and costs are the most important challenges facing the AEC industry. There is a need to include the correct direction to be taken in order to escape these challenges. In addition, laws and regulations in developed countries should be in place to reduce negative attitudes and further boost the constructive approach of the AEC industry.

## 6. Conclusions

This study focuses on the current state-of-the-art research related to DTs in the AEC industry. A mixed systematic review method was adopted in order to provide a mapping of DTs and deeper insights into the research gaps and needs. It was found that the pattern of DTs publications in the AEC industry in the 21st century is strongly comparable to the 20th century. Likewise, the majority of publications in DTs in the AEC industry have also contributed by *Automation in construction*. Furthermore, the United States is the leading source of DTs research in the AEC industry. It was also found that BIM in the AEC industry is considered to be the emerging DTs in all other DTs in the AEC industry. From a theoretical perspective, this research is unique in the following ways: (a) this study presents the first bibliometric-qualitative literature review appraising the state-of-the-art of research on DTs in the AEC industry, (b) the research focuses on the research of DTs in the AEC context, including new advancements, challenges, negative attitudes, and future directions (c) and offering recommendations that provide guidelines on how to address the shortcomings in defining further research. From a practical perspective, this study can help practitioners with a modulated reference point that is easily accessible and grasp the latest techniques and methods of DTs research in the AEC industry.

Despite its contributions, this study has limitations. First, the data is extracted from a Scopus database. Further data will, in future, be collected by integrating data from different databases for quantitative and qualitative analyses (e.g., Google Scholar, Web of Science and so on). Secondly, this research was limited to journal articles only. For these reasons, the results of the research do not fully reflect the entire available literature on DTs in the AEC industry. The limitations listed above provide excellent opportunities for further study, though they should be taken into account when evaluating the results of the research. In future studies, however, data from various sources and a set of parameters may be used for literary impact evaluations, coherence, and linkages to overcome these limitations.

## Figures and Tables

**Figure 1 ijerph-18-06135-f001:**
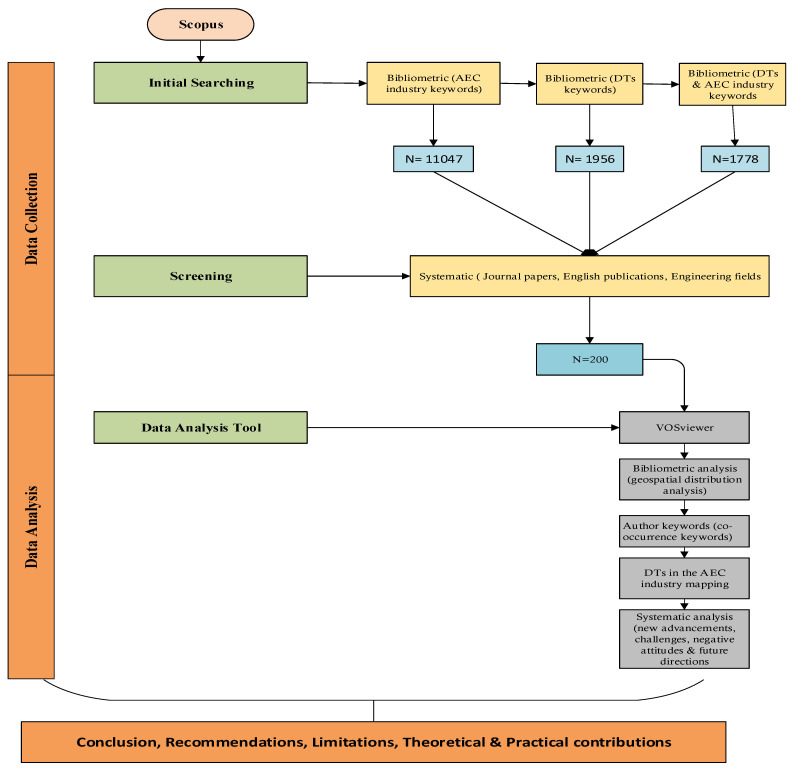
Research flowchart design.

**Figure 2 ijerph-18-06135-f002:**
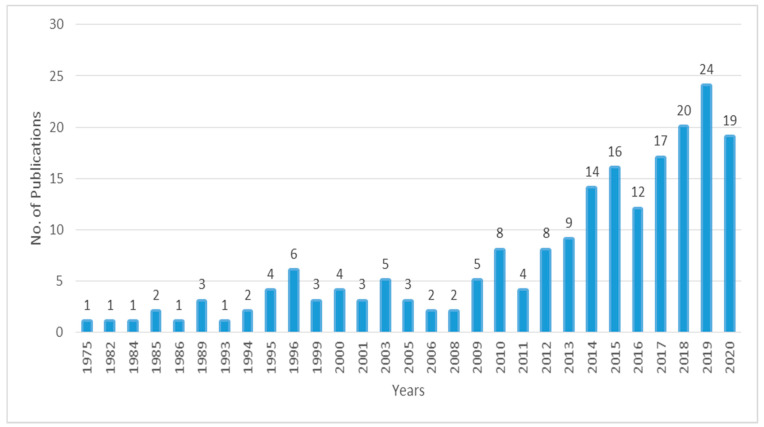
The annual publications trend.

**Figure 3 ijerph-18-06135-f003:**
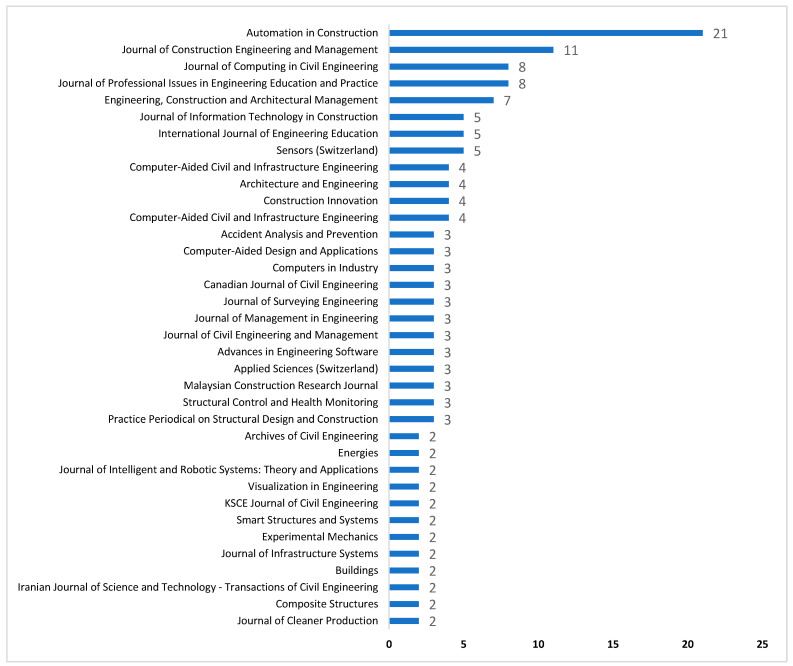
Journal Contributions.

**Figure 4 ijerph-18-06135-f004:**
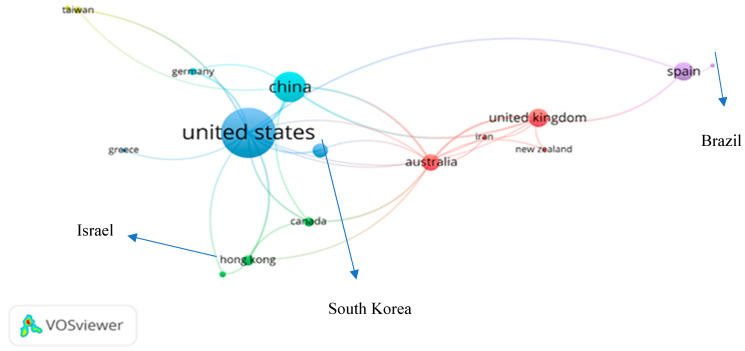
The geospatial collaboration network in research on DTs in the AEC industry.

**Figure 5 ijerph-18-06135-f005:**
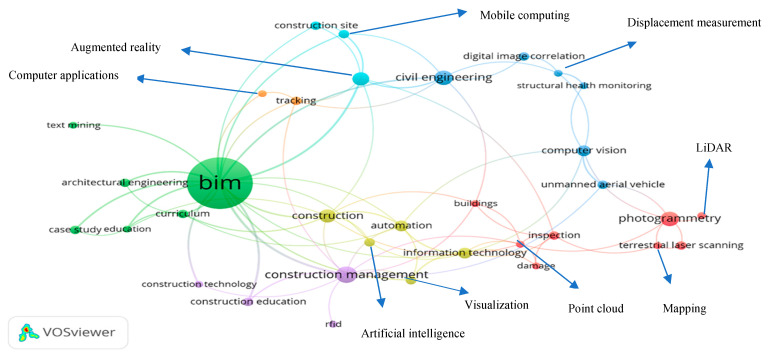
Research interests on DTs in the AEC industry (keywords co-occurrence).

**Figure 6 ijerph-18-06135-f006:**
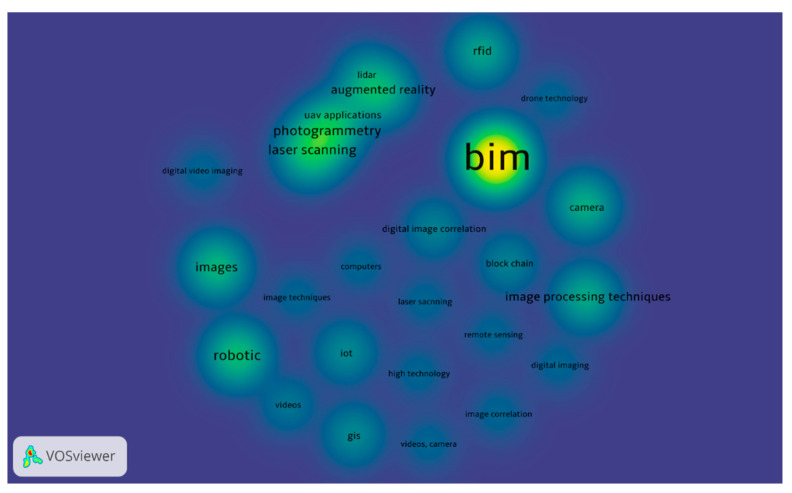
Mapping of DTs in the AEC industry.

**Figure 7 ijerph-18-06135-f007:**
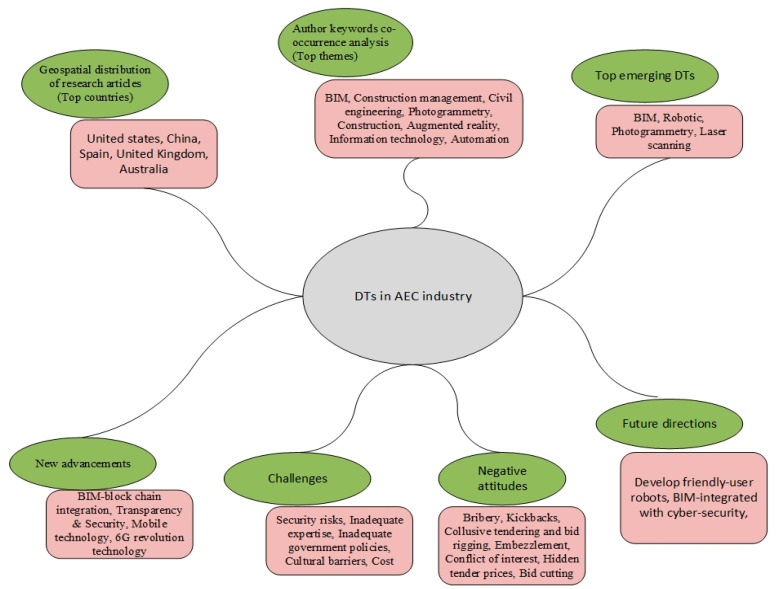
Summary of findings.

**Table 1 ijerph-18-06135-t001:** Literature searching strategies.

Topic	Keywords	Results
the AEC industry	“Digital” OR “technology” OR “technologies” AND “Architectural engineering” OR “Construction engineering” OR “Civil engineering”	11,047
Digital technologies	“BIM” OR “Building information modeling” OR “Building information modelling” OR “AR” OR “VR” OR “RFID” OR “Photogrammetry” OR “Images” OR “Videos” OR “Cameras” OR “laser scanning” OR “3D scanning” OR “Robotics” OR “Block chain”	1956
Digital technologies	“Digital” OR “technology” OR “technologies” AND “Architectural engineering” OR “Construction engineering” OR “Civil engineering” AND “BIM” OR “Building information modeling” OR “Building information modelling” OR “AR” OR “VR” OR “RFID” OR “Photogrammetry” OR “Images” OR “Videos” OR “Cameras” OR “laser scanning” OR “3D scanning” OR “Robotics” OR “Block chain”	1778
Digital technologies and the AEC industry	Screening on the basis of engineering fields, journal papers, unrelated topics, non-English publications	200

**Table 2 ijerph-18-06135-t002:** Geospatial distribution of research articles on DTs in the AEC industry.

Countries	Research Articles	Average Publication Year
United States	60	2010
China	33	2017
Spain	18	2016
United Kingdom	18	2016
Australia	16	2017
South Korea	13	2014
Hong Kong	9	2015
Canada	8	2012
Germany	6	2016
Taiwan	6	2014
Israel	5	2007
Poland	4	2015
Brazil	3	2018
Greece	3	2009
Iran	3	2018
New Zealand	3	2019
Austria	2	2002
Finland	2	2013
Hungary	2	2015
India	2	2020
Italy	2	2020
Singapore	2	2016
Switzerland	2	2007

**Table 3 ijerph-18-06135-t003:** Top keywords of research interest on DTs in the AEC industry.

Keywords	Occurrences	Average Publication Year	Links	Total Links Strength
BIM	60	2017	19	46
Construction management	11	2013	11	16
Civil engineering	9	2017	7	10
Photogrammetry	9	2016	6	6
Construction	8	2010	9	13
Augmented reality	8	2014	7	12
Information technology	6	2015	8	10
Automation	6	2007	7	9
Computer vision	6	2016	6	7
Inspection	4	2015	6	6
Curriculum	4	2011	5	9
Tracking	4	2012	5	5
Visualization	4	2013	4	6
Mobile computing	4	2013	4	6
Construction site	4	2013	3	3
Artificial intelligence	4	2013	3	3
Unmanned aerial vehicle	4	2017	3	4
Architectural engineering	4	2019	2	4
Case study	4	2017	2	4
Construction education	4	2015	2	6
Digital image correlation	4	2018	2	2
Terrestrial laser scanning	4	2016	2	2
Computer applications	3	2014	7	7
Buildings	3	2016	5	5
Displacement measurement	3	2016	4	5
Education	3	2016	4	7
Damage	3	2015	3	3
LiDAR	3	2017	3	3
Mapping	3	2006	2	2
Construction technology	3	2019	2	4
Structural health monitoring	3	2017	2	3
Point cloud	3	2015	1	1
Rfid	3	2012	1	1
Text mining	3	2019	1	2

**Table 4 ijerph-18-06135-t004:** Summary of new advancements towards DTs.

New Advancements	References
Facility management activities	[164,165,166]
Safety Management	[167,168]
Energy management	[169]
Supply chain management	[170,171]
Quality control	[172]
Emergency management	[173,174,175]
Retrofit planning	[136,176]
BIM–block chain integration	[177]
Carbon emissions	[178,179]
Monitoring and Tracking assessment	[165,173,180]
Transparency and Security	[181,182]
Mobile technology	[183]

**Table 5 ijerph-18-06135-t005:** DTS adoption challenges in the AEC industry.

Challenges	References
Inadequate expertise	[193,194]
Inadequate government policies	[195,196]
Cultural barriers	[197]
Cost	[198,199]
Inadequate demand from clients	[200]
Resistance to change	[201,202]
Security risks	[198,202,203]
Inadequate staff	[204,205]
Competing initiatives	[206,207]
Lack of industry standards	[194,208]
Lack of decision support tools	[209,210,211]
Liability concerns	[212,213]

**Table 6 ijerph-18-06135-t006:** Negative attitude towards DTs in the AEC industry.

Negative Attitudes	References
Bribery	[219]
Kickbacks	[220]
Collusive tendering and bid rigging	[221,222]
Embezzlement	[223]
Conflict of interest	[224,225]
Hidden tender prices	[226]
Bid cutting	[221,227]

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
