# Peer review of "Digital Technologies in the Architecture, Engineering and Construction (AEC) Industry—A Bibliometric—Qualitative Literature Review of Research Activities"

_ijerph, 2021, doi:10.3390/ijerph18116135_

Round 1
Reviewer 1 Report
This study explores the state-of-the-art research on DTs in AEC industry by means of a bibliometric-qualitative review method. This research would uncover new knowledge gaps and practical needs in the domain of DTs in AEC industry. This study brings attention to body of knowledge by envisioning trends and patterns by defining key research interests, journals, countries, new advancements, challenges, negative attitudes and future directions towards DTs in AEC industry. This study providing a broad updated review of DTs in AEC literature and lay a foundation for the future researchers, policy makers and practitioners to explore the limitations in future research.
Some problems related to the editing of the article are detected, which must be corrected:
- Line 132. One of the parentheses (.
- Table 3 must appear complete on the same page of the document.
- Figure 6 when using a dark color background, the written text is misread.
- In tables, 4, 5 and 6 all lines must have the same line spacing.
- Line 458, after the quote [181] there is one left over,
- Appointments 64 and 68 match. Authors must delete one of them and correct the numbering of all citations from that point on.
- In the citation [107], you must remove the second surname of the first author.
- Throughout the text, when you start a few sentences, there are extra blank spaces.
Author Response
Dear reviewer 1,
I hope you consider them adequate for this paper to be finally published in IJERPH.
Best regards,
Point 1: Line 132. One of the parentheses (.
Response 1: According to the reviewer’s comments, the point is addressed in line 132.
Point 2: Table 3 must appear complete on the same page of the document.
Response 2: The table 3 is modified accordingly.
Point 3: Figure 6 when using a dark color background, the written text is misread.
Response 3: The figure 6 is the analysis done in VOSviewer, the background colour is built in, we modified it and marked the names in bold letter, now easily readable.
Point 4: In tables, 4, 5 and 6 all lines must have the same line spacing.
Response 4: According to reviewer’s comments, the tables 4,5 and 6 is modified and highlighted in mark changes.
Point 5: Line 458, after the quote [181] there is one left over,
Response 5: According to reviewer’s comments, the line 458 after the quote [181] there is one left over, is modified and highlighted in mark changes.
Point 6: Appointments 64 and 68 match. Authors must delete one of them and correct the numbering of all citations from that point on.
Response 6: According to the reviewer’s comment, 68 is deleted and also correct te numbering of all citations.
Point 7: In the citation [107], you must remove the second surname of the first author.
Response 7: The second surname of the first author has been removed and highlighted in mark changes.
Point 8: Throughout the text, when you start a few sentences, there are extra blank spaces.
Response 8: According to the review’s comment, blank spaces have been removed in second proofread.
Reviewer 2 Report
This manuscript conducted a comprehensive literature review on digital technology in Architecture, Engineering and Construction (AEC) industry. Overall, the topic of this study is interesting, and the structure of manuscript was well organised. Only one concern is about the challenges in this area. I hope the authors can add a section to summarise the challenges in this field and future research directions.
Author Response
Dear reviewer 2,
I hope you consider them adequate for this paper to be finally published in IJERPH.
Best regards,
Point 1: This manuscript conducted a comprehensive literature review on digital technology in Architecture, Engineering and Construction (AEC) industry. Overall, the topic of this study is interesting, and the structure of manuscript was well organised. Only one concern is about the challenges in this area. I hope the authors can add a section to summarise the challenges in this field and future research directions.
Response 1: Thank you for your kind comment and helping to strengthen the manuscript. According to your comment, the challenges and future directions is modified and summarise in table 5.
Reviewer 3 Report
Presenting a literature review on digital technologies in the AEC industry may help to understand their importance. However, the authors must decide whether this is what they mean and if so in what context. In order to emphasize the contribution of the article, and for the purpose of understanding its usefulness, authors are advised to refer to the following comments
1. General comments
Authors are advised to perform thorough proofreading and editing in order to clarify their intentions.
- The title of the article is not understood. It is advisable to emphasize whether the intention is to present the importance of digital technologies in the literature. The reader needs to know exactly what is being considered in the connection between digital technologies and the AEC industry. Benefits? Trends?
- Lines 71. Please explain what the word "tress" means. Try to rephrase the sentence for more understanding
- Line 439. Please explain what is meant by wording "green house buildings".
2. Specific comments
Beyond the need for comprehensive editing of the text, it seems that there is an essential need for a precise definition of the subject, the basic assumptions and the methodology. Presenting them may help refine the meanings of the results, and understand the importance of the research presented.
2.1. Introduction
It is advisable to define in advance what digital technologies are being researched along with the aspects in which they are beneficial. These aspects have been presented in the abstract and should be addressed to allow for consistency in research and understanding.
2.2. Background
- The authors chose to present a historical overview of digital technologies without sufficient reference to the AEC industry that is the subject of the study. It is advisable to reformulate this section.
- It is recommended to present a comprehensive background in relation to the selected digital technologies and to detail in relation to the aspects in which they benefit. For example, according to what is stated in the abstract: "enhanced visualization, better data sharing, a reduction in building waste, increased productivity, sustainable performance and safety improvement".
2.3. Research Methodology
- The method of data collection is vague and may lead to biases and distortions of results.
- Lines 167 - 172. It should be explained why these concepts were chosen and why the specific combinations between them were made. Reference should be made to the use of terms such as "Images" and "cameras". Using arbitrary words, and in the plural form, may ignore many results.
- Bibliometric analysis tools. The use of the chosen tool and how the research was conducted using it should be explained.
2.4. Results and Discussion
- The authors note a comparison between years of publication but do not explain what the results mean in terms of digital technologies and their benefits. It is recommended to list the results according to the aspects presented.
- The authors note the contribution of the journals in relation to digital technologies but also in relation to this there is no reference to the various aspects and the meaning of the results is not understood.
- Geospatial distribution. The diagram must be explained, the thickness of the connections must be visually presented, in order to understand the strong connections. Moreover, the significance of the state division, and the average presented, are not understood. The results need to be explained in terms of the AEC industry in the various aspects.
- Author keywords. The importance of keywords and average presentation are not understood. The results need to be explained in terms of the AEC industry in the various aspects.
- Discussion of clusters. BIM is a prime example of digital technology, especially in the face of results. It is recommended to expand on it, as in relation to other digital technologies, while providing references from the various relevant areas such as: visualization, data sharing, waste, productivity, sustainable execution, green building and improving safety, which the authors mentioned.
Author Response
Dear reviewer 3,
I am enclosing a file with the response to your suggestions.
I hope you consider them adequate for this paper to be finally published in IJERPH.
Best regards,

Round 2
Reviewer 3 Report
In the report on paper the authors were offered to emphasize the contribution of the article, and for the purpose of understanding its usefulness, the authors were advised to address various comments. Given the fact that a topic like digital technologies in the construction industry is not directly related to the Journal of Environmental Research and Public Health, the authors had to explain well the research question and its purpose.
- An example of the importance of making information accessible to readers of this journal is the title of the article, which is general and does not direct the reader to any purpose. What is being explored in digital technologies? For example, research activities in the field.
2.1. The introductory chapter does not bring the journal reader to understand the gaps in the literature that the authors want to explore. The authors ask a number of questions but do not explain why they were chosen and what their importance is.
2.2. It is advisable to present a background relating to the subject of the paper and its purpose together with references.
2.3. Because the use of certain keywords may have an impact on the results of the study, one should examine how they are used, including the use of the plural form versus the singular form which may change the conclusions. The reason for the selected bibliometric tool and how the research was conducted using it must be explained. The references presented in the answer should be presented as part of the explanations.
2.4. It is important to explain what the results of the contribution of journals in relation to digital technologies mean.
Author Response
Dear Reviewer,
Please find attached a file with the response to your suggestions for improvement of this paper. I also attach the updated article following your indications.
I hope that everything is correct for its publication in IJERPH.
Best regards,
